# Targeted Delivery of Nanoparticles to Blood Vessels for the Treatment of Atherosclerosis

**DOI:** 10.3390/biomedicines12071504

**Published:** 2024-07-06

**Authors:** Qiushuo Zong, Chengyi He, Binbin Long, Qingyun Huang, Yunfei Chen, Yiqing Li, Yiping Dang, Chuanqi Cai

**Affiliations:** 1Department of Vascular Surgery, Union Hospital, Tongji Medical College, Huazhong University of Science and Technology, Wuhan 430022, China; u201710323@hust.edu.cn (Q.Z.); animechan@126.com (Y.C.); yiqingli@hust.edu.cn (Y.L.); 2Department of Vascular Surgery, Fujian University of Traditional Chinese Medicine, Fuzhou 350122, China; 2220502015@fjtcm.edu.cn; 3Department of General Surgery, Taihe Hospital Affiliated to Hubei University of Medicine, Shiyan 442099, China; long870209@hotmail.com; 4Department of Cardiothoracic Surgery, The First Hospital of Putian Affiliated to Fujian Medical University, Putian 351106, China; hqyhust@outlook.com

**Keywords:** nanoparticles, atherosclerosis, targeted delivery, responsive nanoparticles, intravascular technology, nano-coated devices, adventitial injection

## Abstract

Atherosclerosis is a common form of cardiovascular disease, which is one of the most prevalent causes of death worldwide, particularly among older individuals. Surgery is the mainstay of treatment for severe stenotic lesions, though the rate of restenosis remains relatively high. Current medication therapy for atherosclerosis has limited efficacy in reversing the formation of atherosclerotic plaques. The search for new drug treatment options is imminent. Some potent medications have shown surprising therapeutic benefits in inhibiting inflammation and endothelial proliferation in plaques. Unfortunately, their use is restricted due to notable dose-dependent systemic side effects or degradation. Nevertheless, with advances in nanotechnology, an increasing number of nano-related medical applications are emerging, such as nano-drug delivery, nano-imaging, nanorobots, and so forth, which allow for restrictions on the use of novel atherosclerotic drugs to be lifted. This paper reviews new perspectives on the targeted delivery of nanoparticles to blood vessels for the treatment of atherosclerosis in both systemic and local drug delivery. In systemic drug delivery, nanoparticles inhibit drug degradation and reduce systemic toxicity through passive and active pathways. To further enhance the precise release of drugs, the localized delivery of nanoparticles can also be accomplished through blood vessel wall injection or using endovascular interventional devices coated with nanoparticles. Overall, nanotechnology holds boundless potential for the diagnosis and treatment of atherosclerotic diseases in the future.

## 1. Introduction

Atherosclerosis (AS) is the primary form of cardiovascular disease, which has been the leading cause of death globally for many years [1,2]. It is typically recognized that oxidized low-density lipoprotein and oxidized phospholipids activate endothelial cells (ECs), causing monocytes to adhere and accumulate beneath the arterial endothelium. Then, the monocytes differentiate into M1-type macrophages that secrete pro-inflammatory factors (TNF-α, IL-1α, IL-1β, IL-6, IL-12, and IL-23, etc.) and cause chronic inflammatory fibroplasia of the vessel wall [3,4]. Subsequently, a portion of the macrophages ingest oxidized LDL and transform into foam cells, while others undergo apoptotic necrosis due to endoplasmic reticulum stress and form the necrotic core of the plaque, collectively resulting in luminal narrowing [5,6]. The necrotic core expands in size with the number of foam cells, and activated macrophages continue releasing matrix-degrading proteases to weaken the fibrous cap covering the plaque, which is made up of smooth muscle and ECs. This makes atherosclerotic lesions more prone to plaque rupture. Furthermore, damage to the local endothelium and tissue may expose tissue factors directly to the blood, which then triggers platelet activation and thrombosis [7,8].

The objective of atherosclerosis medication therapy is to enhance circulation, reduce cholesterol levels, and prevent thrombosis. Anti-inflammatory medication may have modest efficacy when combined with conventional regimens, given that inflammation is a pivotal factor in atherosclerosis [9,10]. Nevertheless, it should be noted that all medication therapy has limitations in treating established plaques. In the case of severe ischemia and high degrees of stenosis, surgical removal of sclerotic plaques or luminal dilatation aided by balloons and stents is advised [11,12]. However, surgical treatment inevitably results in some degree of damage to the vessel wall and has a relatively high restenosis rate. The primary causes of restenosis are the development of thrombosis in the short-term postoperative period and the promotion of intimal hyperplasia that is mediated by a number of inflammation-related mechanisms in the three-month postoperative period [13,14]. Abnormalities in the lipid metabolism and inflammation of the arterial wall are strongly associated with the progression of atherosclerosis and restenosis after surgery. Some medications, such as colchicine, rapamycin, and nucleic acid drugs, possess strong anti-inflammatory, lipid-lowering, or anti-proliferative properties, which means they have great potential in inhibiting atherosclerotic plaque growth and postoperative restenosis. Unfortunately, their utility in the therapy of atherosclerosis is limited by instability or dose-dependent toxicity [15,16,17]. And the introduction of nano drug delivery system (DDS) technologies has shed light on the utilization of these medications. By temporarily isolating the drug from the body’s internal environment during delivery to reduce degradation and avoiding dose-related drug toxicity through effective targeted delivery, these use constraints of these promising drugs can be removed.

In the field of drug delivery, nanoparticles usually refer to solid colloidal particles with diameters between 10 and 1000 nm, in which drugs are adsorbed, dispersed, or encapsulated to form nanospheres and nanocapsules [18,19]. In 2011, nanoparticles were defined by the European Commission as particles that make up 50% or more of the size range of 1 to 100 nm [20]. DDSs have various significant advantages over traditional systemic administration methods. Drug encapsulation or dispersion within nanoparticles minimizes needless degradation. Responsive or targeted DDSs improve local accumulation to increase local drug concentration and reduce the side effects associated with systemic administration; multiple drugs can be delivered by the same nanoparticle carrier for combination therapy to overcome drug resistance [21,22,23]. Furthermore, due to their minute diameter, nanoparticles are more likely to penetrate the body’s defensive barriers, including the blood–brain barrier, mucous membranes, and skin, as well as enter the tissues where they will be absorbed by the cells more efficiently via endocytosis or other mechanisms [24]. Nanoparticle carriers can be broadly divided into two groups: organic and inorganic (Table 1). Polymers, liposomes, and micelles are typical examples of organic compounds, whereas inorganic compounds include silica, metals, and carbon, among others. Liposomes are the most extensively employed nanocarriers for treating atherosclerosis; nevertheless, liposomes have low hydrophilic drug encapsulation and release efficiency. Polymer nanoparticles and micelles are more adaptable in terms of shape, size, release rate, and drug-carrying capacity [25].

Nanoparticles have been the subject of a great deal of research until now. They have a wide application scope, not only for atherosclerosis [26,27] but also for restenosis [28,29,30], cancer [31], aneurysm [32], neointimal hyperplasia [33,34], and thrombosis [35], given their precise targeting and local accumulation ability (Figure 1). According to some recent studies, nanoparticles showed great potential in both clinical diagnosis and therapy for AS [36]. Nevertheless, despite the encouraging outcomes observed in cell and animal studies, only a limited number of these designs have successfully transitioned to clinical trials and, even more rarely, to the market (Table 2). This paper summarizes the progress of nanoparticle targeted therapy for atherosclerosis in terms of two delivery methods—the systemic delivery of drug-loaded nanoparticles via intravenous administration and local delivery, which includes intravascular (via stent or balloon) or perivascular administration to diseased tissue. The objective of this review is to screen and summarize novel or high-performing DDS designs for the treatment of atherosclerosis and to accelerate the translation of more effective therapeutic options into the clinic.

**Table 1 biomedicines-12-01504-t001:** Classification, synthesis, and application of medical nanoparticles.

Category	Material	Nanoparticles		Medical Application	References
		Shape	Synthesis Method		
Polymer	PCL, PLA, PEG, PLGA, chitosan, alginate, gelatin, and albumin, etc.	Spheres, rectangular disks, rods, worms, oblate ellipses, elliptical disks, and circular disks, etc.	Solvent evaporation, salting-out, micro-emulsion, mini-emulsion, dialysis, surfactant-free emulsion, supercritical fluid technology, and interfacial polymerization.	Drug delivery; wound healing; antibacterial agents.	[25,37]
Liposomes	Lipid	Spherical or ellipsoid	Thin-film hydration, ethanol injection, reverse phase evaporation, detergent depletion, etc.	Drug delivery; vaccines; imaging agents; gene delivery.	[38,39]
Non-metal	Carbon, SiO_2_, etc.	Fullerenes, quantum dot, tubes, etc.	Photolithographic techniques, grinding, sputtering, and milling, etc.	Fluorescent probes	[40,41]
Metals and oxides	Iron-oxide, gold, silver, and TiO_2_, etc.	Spherical, rod-like, cage-like, etc.	Chemical reduction, green chemistry, sonochemistry, electrochemistry.	Drug delivery; anemia; imaging agents; gene delivery; molecular diagnosis; bone cements, antibacterial, antiviral, and antifungal agents.	[38,42,43]

Abbreviations: PCL = polycaprolactone; PLA = polylactide; PEG = polyethylene glycol; PLGA = poly lactic-co-glycolic acid.

**Table 2 biomedicines-12-01504-t002:** Clinical trials of nanoparticles for the diagnosis or treatment of atherosclerosis.

NCT Number	Study Title	Study Status	Enroll Number	Interventions	Study Results		Applications
					TAV (Mean ± Standard Deviation, mm^3^)	MACE-Free Survival (%)	
NCT01270139 [44]	Plasmonic Nanophotothermal Therapy of Atherosclerosis	Completed	180	Intervention 1 (procedure): transplantation of NP; Intervention 2 (procedure): transplantation of iron-bearing NP; Intervention 3 (device): stenting	Nano group: 108.2 ± 42.2; Ferro Group: 115.6 ± 64; Stenting Control: 178 ± 52.6	Nano group: 94.3; Ferro Group: 91.4; Stenting Control: 90.5	Stable angina; heart failure; atherosclerosis; multivessel coronary artery disease
NCT00518284 [45]	Prevention of Restenosis Following Revascularization	Terminated	6	Intervention (drug): paclitaxel NP	NA		Vascular disease; peripheral vascular disease
NCT04616872 [46]	Treatment of Patients With Atherosclerotic Disease With Methotrexate-associated with LDL Like Nanoparticles	Unknown	40	Intervention 1 (drug): LDE-methotrexate; Intervention 2 (drug): LDE-placebo	NA		Atherosclerosis; coronary artery disease
NCT04148833 [47]	Treatment of Patients With Atherosclerotic Disease With Paclitaxel-associated with LDL Like Nanoparticles	Unknown	40	Intervention 1 (drug): LDE-paclitaxel; Intervention 2 (drug): LDE-placebo	NA		Atherosclerosis; coronary artery disease
NCT01436123 [48]	Plasmonic Photothermal and Stem Cell Therapy of Atherosclerosis Versus Stenting	Terminated	62	Intervention 1 (device and drug): stenting and micro-infusion of NP; Intervention 2 (device): implantation of everolimus-eluting stent	NA		Atherosclerosis; coronary artery disease
NCT06399328 [49]	Cardiovascular Risk Stratification on the Basis of Surface Enhanced Raman Spectroscopy	Recruiting	220	Intervention (diagnostic test): surface enhanced Raman spectroscopy	NA		Coronary atherosclerosis of native coronary artery

Abbreviation: TAV = total atheroma volume; MACE = major adverse cardiovascular event; NP = nanoparticle.

## 2. Systemic Delivery of Drug-Loaded Nanoparticles

Nanoparticles can be delivered systemically via both passive and active pathways (Figure 2). The systemic delivery could be influenced by interactions between nanoparticles and plasma proteins, blood cells, and ECs, as well as the excretion and clearance of renal, hepatic, and mononuclear phagocyte systems (MPSs) [50]. In total, 30–99% of systemically delivered nanoparticles aggregate in the liver, reducing the efficiency of targeted drug delivery while simultaneously increasing the risk of hepatotoxicity [51]. Therefore, when designing nanoparticles for systemic administration, two essential parameters that influence delivery efficiency should be considered: nanoparticle circulation time and the efficient binding of the nanoparticles to the target. Enhancements to the physical properties of the nanoparticles, such as their size, shape, polymer length, and surface activity, can be improved to maximize their cycle time and likelihood of binding to a specific target.

### 2.1. Targeted Nanoparticle Delivery Based on Physicochemical Properties

Magnetism and the enhanced permeation and retention (EPR) effect of nanoparticles are widely used as a passive accumulation approach. Furthermore, the sensitivity of nanoparticles to shear, heat, light, ultrasound, thermal irradiation, and changes in the pH of the microenvironment cause the breakdown of nanoparticles at the target location, releasing the drug for therapeutic purposes and allowing their distinctive response functions [52] (Table 3).

Magnetic fields can be utilized to direct the local aggregation and distribution of magnetic nanoparticles (MNPs) to transport ECs to improve vascular function. Vosen et al. loaded ECs overexpressing endothelial nitric oxide synthase (eNOS) using a combination of lentiviral vectors and MNPs, localizing at the site of endothelial damage following the mechanical removal of plaque, to enhance vascular function by endothelializing the artery radially and symmetrically [53]. Polyak et al. discovered, in a rat carotid stent angioplasty model, that the magnetically assisted administration of ECs provided considerable protection against mechanical-injury-induced ISR [55]. Moreover, polylactide-based MNPs allow for non-invasive monitoring of endothelial cell location and proliferation by the addition of BODIPY 558/568 labeling and exhibited enhanced magnetic responsiveness, resistance to cryopreservation, and rapid expansion [54]. Furthermore, the EPR effect has also been used to treat AS using nano DDSs, and its targeting efficiency has been verified. Systemic passive administration via the EPR effect is the most extensively researched targeted method for nano DDSs in the context of cancer [58]. The EPR effect is also evident locally in atherosclerotic lesions, where the newborn immature vascular ECs have not formed unbroken tight connections with one other and are generally more permeable than normal arteries [59,60]. Utilizing ultrastructural and en face plaque imaging, Beldman et al. displayed the localization of hyaluronan nanoparticles in both early and advanced atherosclerotic lesions. Their findings demonstrated that atherosclerotic plaques had significantly lower endothelial connectivity continuity than the normal vascular endothelium, which may have contributed to the accumulation of nanoparticles at the lesion site and served as the primary route for their transportation to the sub-endothelium [59].

Alterations in the hemodynamics and microenvironment surrounding the lesion may serve as a response approach for nanoparticle-targeted treatment. Changes in blood flow washout patterns in the bends and branches of arteries reduce shear stress and blood flow velocity, whereas low shear stress promotes the development of atherosclerotic plaque [61,62]. Following the formation of arterial plaque, the high shear stress at the stenosis and the low shear stress at the stenosis’s distal exit generate turbulence; some shear unstable nanoparticles take advantage of this turbulence to induce the disintegration of the intact structure of the particles and leakage of the drug, completing the mission of transporting the drug to the location of the lesion. Localized inflammation during balloon angioplasty is related to elevated reactive oxygen species (ROS) levels and a weakly acidic environment. These inflammatory mechanisms are intimately associated with the development of restenosis [63,64]. Feng et al. prepared pH- or ROS-responsive nanoparticles that encapsulated rapamycin to inhibit intimal hyperplasia in a rat model of arterial restenosis using a nanoprecipitation technique [56]. During the course of atherosclerotic disease, ECs and smooth muscle cells of the lesioned artery wall exhibit oxidative stress as a result of NAPDH up-regulation and the accompanying ROS release [65]. Thus, in addition to reducing restenosis, weakly acidic and ROS-responsive nanoparticles may have potential applications in slowing the progression of atherosclerotic disease. 

Combining accumulation strategies with characteristic responses allows for the creation of novel smart nano DDSs with efficient targeting to raise local drug concentration and improve therapeutic efficacy. Shen et al. created a drug-carrying system that is sensitive to high shear stress (100 dynes/cm^2^) at plaques and especially responds to the ROS microenvironment. Self-assembled cross-linked polyethyleneimine nanoparticles loaded with simvastatinic acid in erythrocytes enabled the sustained release of ROS and lowered lipid levels [57].

### 2.2. Targeted Drug Delivery Based on Specific Markers

In general, nanoparticles that are selectively targeted based on ligand–receptor interactions are more potent than those targeted only based on their physical characteristics. Therefore, abnormally high expression molecules associated with AS are frequently utilized as targets. For example, αvβ3-integrin expression is significantly higher in AS than in normal arteries, and this makes it a potential biomarker for detecting for damaged vessel detection and a specific target for nanomedicines [66,67]. Similarly, molecules highly expressed in damaged or immature endothelial cells, aggregated macrophages, or exposed vessel wall collagen inside the plaque are specific targets for nanomedicine delivery in AS (Table 4).

#### 2.2.1. Targeting Damaged or Immature ECs

In the initial phase of atherosclerotic plaque formation, aberrant lipid metabolism, inflammatory response, oxidative stress, and blood flow disturbances trigger EC activation and dysfunction. This leads to the expression of numerous adhesion molecules and inflammatory cell chemokines, including E-selectin, intercellular adhesion molecule 1 (ICAM-1), vascular cell adhesion molecule 1 (VCAM-1), and monocyte chemoattractant protein 1 (MCP-1) [79]. Phospholipid molecules modified with the VCAM-1 target VHP peptide, ultra-small paramagnetic iron oxides, and rapamycin showed therapeutic efficacy equivalent to that of low-dose rapamycin and allowed for the visualization of early atherosclerotic lesions [68]. Liposome nanoparticles loaded with cyclopentenone prostaglandin and VCAM-1 antibodies specifically targeted damaged arterial ECs in atherosclerotic mice and exhibited anti-inflammatory, apoptosis-promoting, anti-adipogenic, and cytoprotective properties similar to heat shock proteins. This resulted in reduced lipid accumulation in the arterial wall [69]. Additionally, Tang et al. discovered that encapsulated colchicine can inhibit the progression of atherosclerotic plaques through blocking the NF-κB/NLRP3 pathway. They also found that poly lactic-co-glycolic acid (PLGA) nanoparticles functionalized with polyethylene glycol (PEG) may prevent MPS scavenging, prolong cycle time, and boost the probabilities of binding VHPK peptide with VCAM-1 on inflammatory ECs [15].

ανβ3-integrin is one of the common specific targets of neovascular ECs. Angiogenesis inside the plaque is a sign of plaque instability, and avoiding neovascularization is a successful strategy for slowing the progression of early plaques to vulnerable plaques. Combining ανβ3-targeted paramagnetic fumagillin nanoparticles with atorvastatin showed antithrombotic and plaque-stabilizing effects [70]. CD31 antibody staining indicated a correlation between its therapeutic effect and the suppression of intraplaque angiogenesis [71]. Meanwhile, the transition to vulnerable plaques in AS are encouraged to form by neutrophil extracellular traps, which are formed through the assembly of cytoplasmic and granular proteins on stents of de-concentrated stained monomers with the aid of neutrophil elastase [80]. In order to overcome the short half-life and lack of specific targeting of sivelestat (SVT), a competitive inhibitor of neutrophil elastase, Shi et al. developed the SVT liposome modified with cRGD peptide, which slows the progression of atherosclerosis by stabilizing and reducing the plaque area [72].

#### 2.2.2. Targeting Exposed Collagen

The extracellular matrix of the artery wall contains a large quantity of collagen IV; however, ECs normally separate collagen and the blood. The quantity of collagen IV in the artery wall rises with the formation of atherosclerotic plaques, and collagen IV becomes more readily exposed following endothelial damage [81]. 

Several studies have demonstrated that collagen IV exposed in plaques is an effective target for drug-loaded nanoparticles. Fredman et al. encapsulated a small fragment of annexin A1 (Ac2-26) into collagen IV-binding peptide-modified polymer nanoparticles to reduce plaque endothelial cell oxidative stress and increase the interleukin-10 (IL-10) level, thereby stabilizing plaques through inhibiting inflammatory progression [73]. Following angioplasty, mechanical disruption to the endothelium layer causes collagen exposure, which promotes platelet adhesion and activation and sets off an inflammatory chain reaction that finally leads to restenosis. Anti-inflammatory peptide loading via collagen-targeted sulfated poly(N-isopropylacrylamide)-modified nanoparticles allows for binding to exposed collagen to reduce platelet adhesion. Meanwhile, the release of anti-inflammatory peptides reduces inflammation of endothelial and smooth muscle cells and promotes endothelialization to inhibit postoperative restenosis [74]. 

The effectiveness with which modified peptides target plaques varies depending on the target. Kim et al. conducted a systematical comparison in an Apo E^-/-^ mouse model and discovered that cRGD targeted early atherosclerotic plaques better than collagen IV-targeted peptides with the identical, conditionally modified iron oxide nanoparticles [82]. To optimize drug delivery efficacy, future drug-loaded nanoparticle assemblies should include not only the affinity differences between various targets but also the selection of ligands with higher affinities for plaques at the different stages of lesions.

#### 2.2.3. Targeting Aggregated Macrophages in Plaques

Macrophages are crucial to the course of atherosclerotic disease. As discussed in Section 2.2.1, ECs undergo biochemical and functional alterations in response to various stimuli throughout the early stages of atherosclerosis. These changes direct monocytes toward chemotaxis, adhesion, infiltration of the local sick arterial wall, and macrophage differentiation, which leads to the creation of the first plaque. The amount of macrophages aggregated inside the plaque promotes endothelial oxidative stress and inflammatory response, as well as the local infiltration of inflammatory cells and plaque expansion [83]. On the surface of macrophages localized to atherosclerotic lesions, an increase in mannose receptor (CD206) has been measured, which is often used as a target for macrophages in plaques for nanoparticle design. In addition, monocytes differentiate into macrophages, with an increased expression of scavenger receptors on the cell membrane, including CD-36, SR-BI, and MSR-A [84,85,86]. Oxidized phosphatidylcholine (oxPC) is abundant on the surface of oxidized low-density lipoprotein (oxLDL) and is responsible for binding to macrophages’ CD36 receptor in atherosclerosis. OxPC-modified liposomes showed a significant affinity for the endothelial macrophage CD36 receptor in the THP-1 cell line, as well as aortic lesions in LDL receptor-deficient (LDLr^-/-^) mice [75]. 

Nucleic-acid-loaded nanoparticles can be utilized to apply tailored therapies by more accurately suppressing the activation of aberrant disease-related molecules or pathways, demonstrating significant therapeutic potential. Nanoparticles with siRNAs loaded with macrophage receptor sta-bilin-2 binding peptide sequences inhibit Ca^2+^/calmodulin-dependent protein kinase γ (CaMKIIγ) activity in diseased macrophages. This is achieved through self-assembly, the down-regulation of effervescent receptor MerTK expression, and late plaque stability. Their performance was evaluated in an atherosclerotic mouse model, demonstrating exceptional targeting efficiency, biocompatibility, and therapeutic effectiveness [76]. Bai et al. constructed three-dimensional spherical nucleic acid nanostructures using oligonucleotide RNA-146a, which naturally targets class A scavenger receptors on plaque macrophages. This modulating of the NF-κB pathway, linked to vascular inflammation and the immune response without viral transfection, can effectively reduce and stabilize plaques [77].

#### 2.2.4. Other Targeting Strategies

The key issue to be addressed in the field of nano DDSs is how to improve the targeting efficiency of nanoparticles so that pharmaceuticals may be delivered to lesions with maximum effectiveness. As the progression of atherosclerotic lesions is strongly correlated with the functional changes in ECs and macrophages, as well as the cellular interactions between them, in addition to nanoparticles that individually target highly expressed molecules on ECs and macrophages, therapeutic modes that target both types of cells are being actively investigated and show initial signs of efficiency. The core–shell nanoplatform developed by Zhao et al. could target ECs and macrophages in atherosclerotic lesions by sequentially releasing atorvastatin and siRNA, targeting the key protein molecule for lipid transport between ECs and macrophages, lectin-like oxidized low-density lipoprotein receptor-1 (LOX-1). The plaque size regressed by 39%, and lipid accumulation was decreased by 63% in comparison to the baseline group [78].

Cell membrane-coated nanoparticles (CNPs) have lately received increased attention for their potential use in nano DDSs. Synthesized nanoparticles are encapsulated in separated cell membranes, keeping the membrane’s native components and signal recognition molecules, which can significantly increase biocompatibility and circumvent MPS scavenging to lengthen cycle time [87]. The majority of the cell membranes utilized in CNPs to treat atherosclerosis are derived from erythrocytes, macrophages, and platelets. Furthermore, the particular targeting potential of CNPs is significantly increased by utilizing the intrinsic affinity of platelets and macrophages for plaque and inflammatory homing effects [88]. Song et al. created platelet-membrane-coated nanoparticles that could efficiently target atherosclerotic plaques, and these displayed 4.98-fold greater radiant efficiency than control nanoparticles [89]. Chen et al. created platelet-membrane-coated mesoporous silicon nanoparticles that delivered anti-CD47 antibodies. Platelet membranes serve to avoid immune identification and target atherosclerotic plaques, whereas the CD47 antibody allows for the normal removal of necrotic cells, stabilizes plaques, and lowers platelet activity, hence reducing the risk of thrombosis [90]. Platelet-membrane-coated nanoparticles can target subclinical sections of arteries at risk of plaque development, as well as actual atherosclerotic plaques. This has therapeutic implications for the prevention of early occult plaques and plaque development [91]. In in vitro experiments and atherosclerotic mouse models, macrophage-membrane-coated biomimetic nanoparticles loaded with anti-inflammatory and anti-proliferative drugs like rapamycin and colchicine demonstrated good therapeutic efficacy and biocompatibility, confirming their excellent targeting properties based on macrophage homing action [17,92]. Modifying macrophage membrane coatings with CD47 and α4/β1-integrin decreased MPS clearance of nanoparticles through immune evasion. Additionally, α4/β1-integrin increased the targeting of inflammatory ECs in plaques [17]. Liu reported a macrophage-membrane-encapsulated ROS-responsive nanoparticle, which was validated in ApoE^-/-^ mice in terms of its good targeting and release of loaded drugs in response to the high-level ROS environment at the lesion. Furthermore, the macrophage membrane encapsulation reduced the level of inflammation [93].

## 3. Localized Direct Delivery of Drug-Loaded Nanoparticles in the Vessel Wall

Unlike systemic delivery, interventional devices can administer medications directly to the site of the lesion, resulting in better local drug retention. Restenosis is now the most prevalent cause of vascular dysfunction following stent placement or balloon angioplasty. Thrombosis and stenosis are induced by excessive proliferation of endothelial and smooth muscle cells, resulting in early mechanical damage to the endothelium and arterial wall [94]. Antiproliferative drug-eluting stents (DESs) and drug-coated balloons (DCBs) are now being employed in clinical settings. The first generation of DESs, laden with sirolimus and paclitaxel, outperformed bare stents in suppressing restenosis. Second-generation DESs, loaded with everolimus and zotarolimus, are currently being assessed in terms of safety and efficacy [95,96]. Nevertheless, it was revealed that the pure drug coating had a poor local adhesion rate, and the medication’s potency was only sustained for a short period. Drugs that spread throughout the body in the bloodstream have a risk of causing adverse reactions; for instance, anticoagulants are prone to causing bleeding, and antiproliferative drugs are potentially carcinogenic [97]. Using nanoparticles as coatings for interventional devices not only enhances the absorption of drugs by improving endocytosis and extending the period of effectiveness but also reduces the risk of drug diffusion and raises local adhesion stability through target modification [98]. Localized delivery of nanoparticles to the vessel wall can be roughly separated into two routes: intimal and media-adventitial delivery (Figure 3).

### 3.1. Intimal Route Delivery

The delivery of nanoparticles to the intima of the vessel wall is mainly based on conventional nano-coated stents or balloons (Table 5). Nanoparticle-eluting stents exhibit greater absorption rates and a longer effective medication action period compared to drug-impregnated stents. The implantation of cationic NP-eluting stents encapsulated with the fluorescent marker FITC, which was accomplished using a novel cationic electrodeposition coating technique, resulted in significant FITC fluorescence in the intima and mid-layer of the vessel wall of stented segments for up to 4 weeks without exacerbating injury or inflammation. By contrast, no significant FITC fluorescence was found in arteries implanted with polymer-based FITC-eluting stents or bare metal stents [100]. Sirolimus was encapsulated in polyester-based polymers to form 100–400 nm nanoparticles, which were delivered to the coronary arteries of a pig model through a porous balloon. At 26 days after implantation, the sirolimus concentration remained higher than the target therapeutic level (1 ng/mg) with a relative reduction in stenosis, and without lengthy retention in the rest of the body [101]. The distribution and diffusion properties of 400 nm liposomes used for delivering sirolimus were investigated during rabbit iliac artery balloon angioplasty. The findings revealed that liposomes were evenly distributed throughout the artery wall and tended to spread progressively from the intima to the adventitia [102].

In addition to incorporating standard anti-inflammatory and anti-proliferative chemical nanoparticles, gene-eluting stents and TiO_2_ nanotube-coated stents have demonstrated exceptional stenosis resistance. A miR-126-dsRNA-containing nanoparticle (miR-126-NP) inhibited insulin receptor substrate 1 (IRS-1) to reduce vascular smooth muscle cell proliferation and migration. In a rabbit restenosis model, miR-126-NP-coupled stents substantially prevented intimal development [103]. When a titanium bare stent with surface-deposited non-drug-loaded TiO_2_ nanotubular arrays was implanted into a rabbit iliofemoral artery hyperinflation model, the extracellular matrix secreted during functional endothelial recovery integrated into the surface of the TiO_2_ nanotubular arrays, and the stenosis rate was reduced by 30% compared with the control group after 28 days [104]. TiO_2_ nanotubes coated with Ag nanoparticles have bactericidal activity due to the delayed release of Ag+, and UV irradiation increases their anticoagulant capability. They offer considerable promise for implanted device applications in the cardiovascular system, since they specifically decrease smooth muscle cell proliferation and macrophage adhesion without delaying endothelialization and lowering inflammatory responses and proliferation [105,106].

It is also worth noting that the sequential release of antithrombotic and antiproliferative nanoparticle-coated stents inhibited the antiproliferative medicines’ effect on endothelialization. Du et al. created a core–shell structure coat made up of the platelet glycoprotein IIb/IIIa receptor monoclonal antibody SZ-21 wrapped around a core of the antiproliferative chemical docetaxel. It was sprayed to the surface of a stainless-steel stent using coaxial electrospray to enable sequential drug release. In a pig coronary artery model, the coating was shown to promote endothelialization while inhibiting neoplastic endothelial growth [107].

Implantable cardiovascular devices are generally coated with nanoparticles through spraying, impregnation, cation electrodeposition, and electrostatic adsorption. Iyer et al. assessed the efficacy of drug transfer to the arterial wall from DCBs and discovered that the acrylic-based hydrogel coating had the highest efficacy of drug transfer (95%), significantly greater than that of the impregnation coating (20%) and electrostatic coating (60%) techniques under flow conditions [108].

### 3.2. Tunica Media and Adventitial Route Delivery

In a balloon injury model, vascular epithelial cells differentiate into diverse phenotypes in response to vascular damage and inflammation, with fibroblasts changing into smooth muscle-like cells and leading to stenosis [109]. Localized gene transfer from the adventitia to the intima in the vessel wall might have a therapeutic effect on decreasing restenosis and vascular remodeling [110]. Drug administration from the vascular lumen may impede enough drug diffusion into the media and adventitia to suppress smooth muscle cell and fibroblast proliferation, as well as delay endothelialization and impair long-term lumen patency. In light of this difficulty, medication administration via microneedle to the tunica media or the adventitial route is a comprehensive method for both sides (Table 6). Lee et al. added 200 μm long microneedles (MNs) to the balloon using conformal transfer molding and UV curing, allowing for medication loading. Two PVA scaffolds with variable elastic moduli (52 and 183 kPa) were employed for in vitro release experiments. The MNs could be penetrated to a depth that would not injure the media layer but would sufficiently overcome the endothelium (penetration depths between 10 μm and 500 μm) [111]. A bullfrog catheter (Mercator MedSystems, Emeryville, California) is a micro-infusion device that consists of a balloon and a drug-injection component that penetrates the mid-membrane and epithelium to deliver the drug and, unlike drug coatings, has a flexible regimen for injecting the drug that can be adapted as needed [112]. Ang et al. demonstrated the sustained release of sirolimus in the artery for 28 days and a reduction in the inflammatory response in the damaged vessel by injecting sirolimus nanoliposomes via bullfrog catheter into the vessel wall of a porcine femoral artery balloon injury model, resulting in a significant decrease in the rate of luminal stenosis and neointimal area when compared to the control group [113]. In a rat aortic wire injury model, it has been validated that adventitial injection of hyaluronic acid/sodium alginate (HA/SA) hydrogel loaded with PLGA rapamycin nanoparticles inhibited proliferation and inflammatory cell aggregation [114]. In addition, the rapamycin-loaded PLGA nanoparticle/pluronic gel system was delivered through the outer membrane to reduce phosphorylation of S6 kinase (S6K1), which significantly inhibited smooth muscle cell proliferation at both 14 and 28 days, whereas the inhibitory effect of rapamycin applied alone through the outer membrane re-bounded at 28 days [115].

## 4. Discussion and Prospects

Nanoparticles have steadily garnered several medicinal applications over several decades of development, including medication delivery, bio-imaging, and regenerative medicine. However, they encounter a variety of problems related to aspects such as biosafety, biodistribution and clearance, and drug loading and release [116]. The majority of conclusions regarding the toxicity, distribution, and metabolism of nanoparticles are based on the findings of short-term exposure experiments, and the majority of pharmacokinetic assessments of drug-loaded nanoparticles focus on the delivered drug rather than the delivery vehicle. Consequently, one of the causes for insufficient safety evaluations of nanoparticles is a lack of studies on the carrier’s metabolism and excretion, which also serves as a substantial obstacle to nanoparticle clinical translation. Several studies have demonstrated that the accumulation of nanoparticles in various organs might produce an inflammatory response and that the distribution patterns of these particles within distinct organs may be more directly related to their size [117,118,119]. The degree of biocompatibility of a nanoparticle is now recognized to be determined by its inherent physical and chemical features, such as size, shape, surface characteristics, and the human environment with which it interacts [120]. The nanoparticles’ surface is the interface that comes into contact with the human body and plays an important role in mediating interactions of nanoparticles with blood components or cells. These interactions might take the form of cellular physiological function control or toxicity caused by blood incompatibility, cell membrane disruption, immunological responses, accumulation in organs and tissues, and so on. Amir et al. noted mechanisms for nanoparticle damage to cell membranes such as membrane perturbation, oxidative stress signaling, signaling molecule destruction, and genotoxicity. Due to its strong association with cellular internalization efficiency, the surface charge of nanoparticles may be one of the most crucial surface parameters influencing their toxicity [121]. Mokshada et al. discovered that after intravenous injection, the nanoparticle biodistribution coefficient (NBC) was highest in the liver and spleen, with 17.56% ID/g (for nanoparticles per gram of tissue rather than payload) and 12.1% ID/g, respectively. By contrast, other tissues (heart, lung, brain, kidney, stomach, colon, pancreas, bone, skin, and muscle) had NBC levels less than 5% ID/g. Furthermore, based on this nanoparticle distribution data, they built a computer model to simulate the distribution and dynamics of nanoparticles of various materials and qualities, which could be beneficial for future evaluations of nanoparticle distribution features [122]. Beyond the ordinary nano DDSs, the advent of nanorobots offers new opportunities for the treatment of atherosclerosis. Nanorobots are tiny machines that can transform various energy sources into mechanical forces and perform medical procedures. Nanorobots could be sent directly to the lesion site for precise drug delivery or direct physical intervention, making them ideal for drug delivery, tumor diagnostics, targeted therapy, minimally invasive surgery, and other applications. Nanorobots might be developed to recognize and adhere to atherosclerotic plaques, releasing medications in the same way as typical drug-loaded nanoparticles do, but with higher precision and efficiency. Additionally, nanorobots could be utilized to remove plaque from blood vessels and restore vascular patency [123] (Figure 4).

This paper investigated the use of nanoparticles in the management of atherosclerotic disease; however, nanoparticles have several notable applications in the diagnosis of atherosclerosis, including the identification of vulnerable plaques and the use of nanoparticles with both therapeutic and diagnostic functions [125,126]. Nonetheless, even if the number of nanoparticle kinds is rapidly increasing, very few of them have gone to preclinical or clinical phases, and the majority of research has only formed conclusions from tests conducted in rodents (mice, rats) or other mammals (rabbits, pigs, dogs, and primates) [127]. Numerous nanoparticles have produced fairly acceptable results in cell or animal experiments; nonetheless, the evidence about their toxicity, biocompatibility, metabolic breakdown pathways, and efficacy is insufficient due to small sample sizes. On the other hand, the features of several batches of lab-prepared nanoparticles may differ, and evaluating different batches of nanoparticles may yield a more stable evaluation result for a thorough understanding of nanoparticle attributes. Furthermore, the majority of the experimental research examined in this work yielded favorable outcomes, but the negative impacts or qualities of nanoparticles that need to be improved are hardly discussed. A multidimensional focus on all elements of nanoparticle properties, as well as a combination of all of the good features of nanoparticles, are important concerns that must be addressed in future nanoparticle design.

## 5. Conclusions

Overall, as nanomedicine advances, the possibilities for the clinical translation of nanotechnology in disease diagnosis and therapy tend to be quite promising. DDSs have alleviated numerous constraints in the use of medications in the treatment of atherosclerosis, allowing for more accurate targeting of lesions and giving new directions for the development of future pharmacological treatments and interventions. To improve the therapeutic efficiency of drug-loaded nanoparticles, more efficient targeting strategies, potent therapeutic agents, and superior fabrication procedures must be investigated and tested in order to provide more therapy choices for atherosclerosis.

## Figures and Tables

**Figure 1 biomedicines-12-01504-f001:**
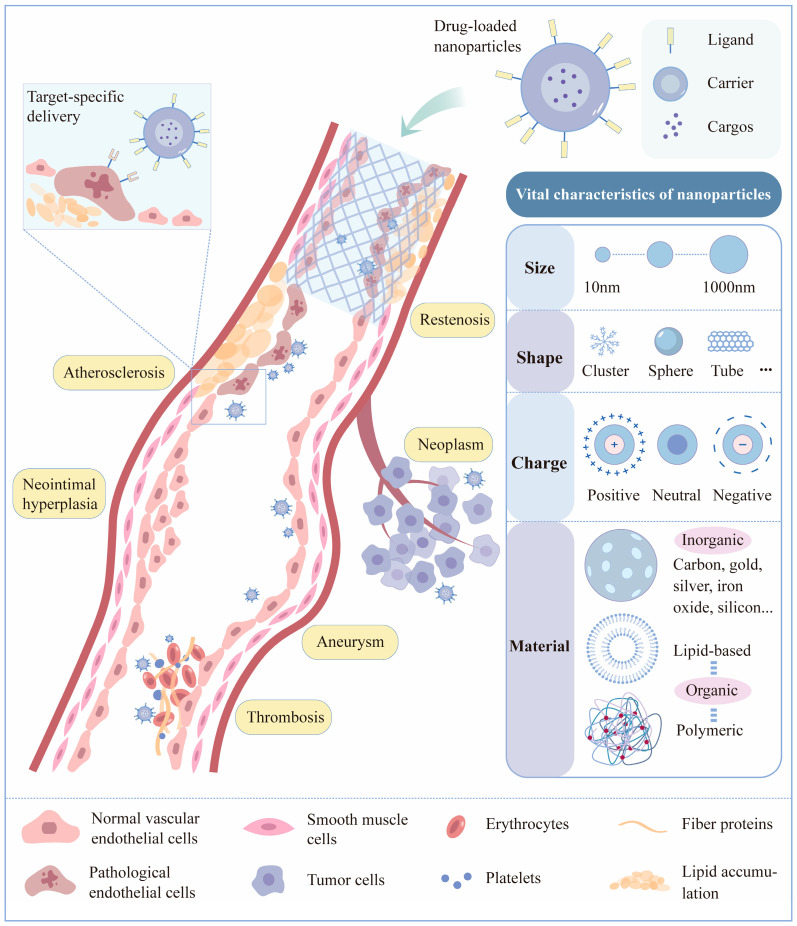
Vital characteristics and applications of nanoparticles. Drug-loaded nanoparticles have numerous medicinal biomedical applications, including in atherosclerosis [26,27], restenosis [28,29,30], neoplasm [31], aneurysm [32], neointimal hyperplasia [33,34], and thrombosis [35]. Their characteristics, including size, shape, surface charge, and material, are closely related to nanoparticle function type. These characteristics determine their cycle time and metabolic distribution after entering the human body and therefore also correlate, to some extent, with the targeting efficiency and organ toxicity of the nanoparticles.

**Figure 2 biomedicines-12-01504-f002:**
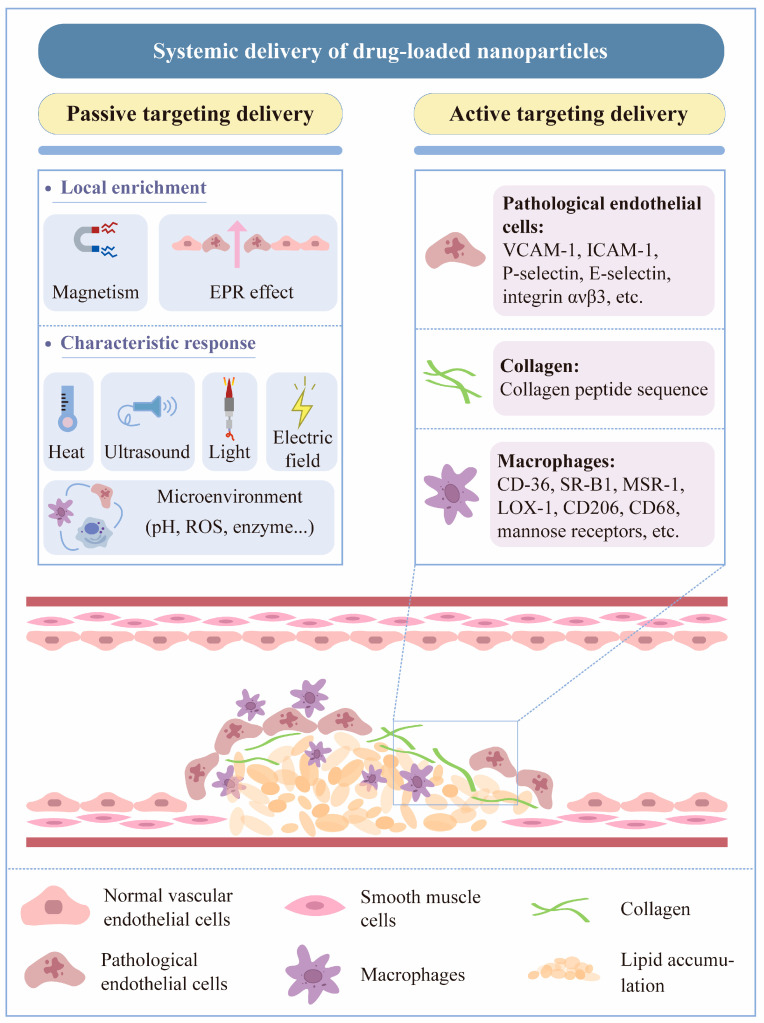
Systemic delivery of drug-loaded nanoparticles. Nanoparticles given systemically, i.e., intravenously, can be targeted to a specific site via two pathways: passive targeting and active targeting. Passive targeting can be achieved by local accumulation through the magnetic or enhanced permeation and retention (EPR) effect of nanoparticles or by designing nanoparticles that exhibit characteristic responses to heat, ultrasound, light, electric fields, or local microenvironments, releasing the loaded drug to the target lesion site [52]. By contrast, nanoparticles for active targeted delivery rely on receptor–ligand interactions to more accurately target a variety of differently expressed molecules in the lesion.

**Figure 3 biomedicines-12-01504-f003:**
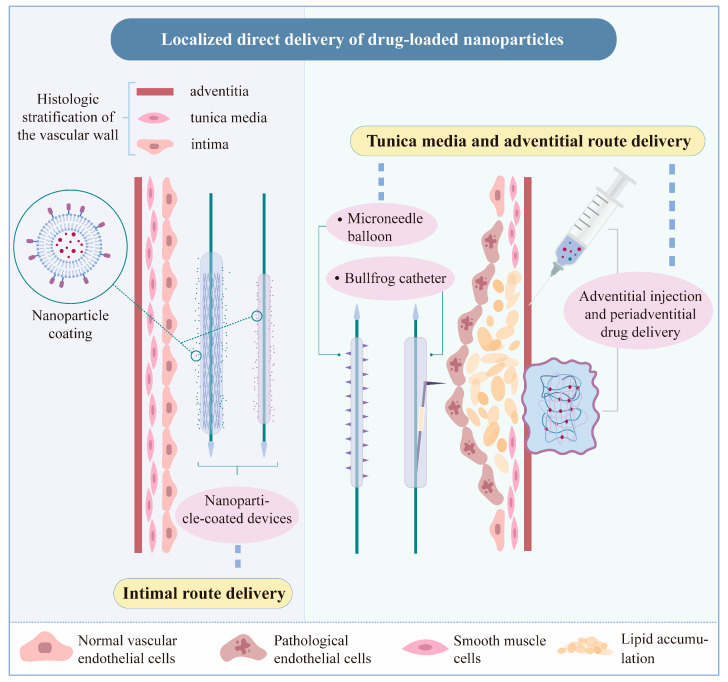
Localized direct delivery of drug-loaded nanoparticles. The vessel wall structure can be divided into three parts: intima, media, and adventitia [99]. The most common method of delivering nanoparticles locally to the vessel wall is based on regular nano-coated devices, such as nano-coated balloons or stents, which transfer nanoparticles to the intima when the device reaches the vessel intima. Furthermore, nanoparticles can be injected into the media or adventitia of the vessel wall from or out of the vessel lumen.

**Figure 4 biomedicines-12-01504-f004:**
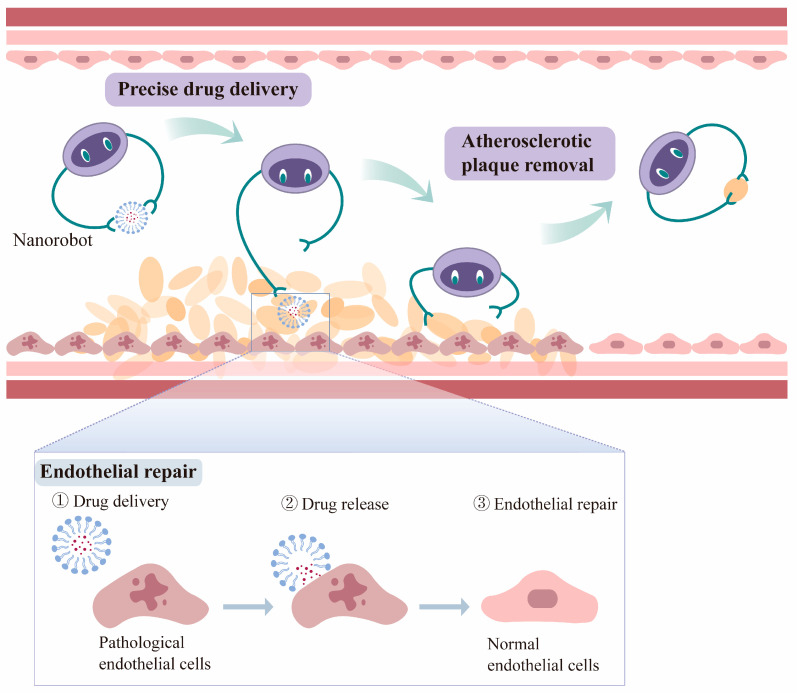
Application of nanorobots in atherosclerosis. The nanorobots can move to the lesion site under manipulation, delivering drugs more precisely for endothelial repair and performing “minimally invasive surgery” to remove atherosclerotic plaque [124].

**Table 3 biomedicines-12-01504-t003:** Nanoparticles based on passive delivery (enrichment or characteristic response).

Nanocarrier	Cargo	Accumulation or Response Strategy	Half-Life (h)	Experiments			References
				Models	Sample Size *	Outcomes	
Iron	eNOS-overexpressing cells	magnetism	NA	eNOS-/- mice; bovine artery ECs	n = 3	At least half of the lumen is covered with ECs	[53]
PLA/magnetite	MNPs functionalized ECs	magnetism	NA	Lewis rats; rat aortic ECs	n = 10	The fastest rapid proliferation and functionalization of ECs	[54]
PLA/magnetite	MNPs functionalized ECs	magnetism	NA	Rat carotid artery stent angioplasty model; rat aortic endothelial cell	n = 26	1.7-fold less reduction in lumen diameter	[55]
β-cyclodextrin	Sirolimus	ROS or pH	NA	Rat carotid artery balloon injury model; rat VSMCs	n = 12	Decrease artery intima–media ratio	[56]
Polyethyleneimine	Simvastatin acid	Shear stress and ROS	11.7 ± 1.2	Rabbit FeCl_3_ thrombosis model	n = 12	The lowest amount of thrombosis	[57]

Abbreviations: PLA = polylactide; eNOS = endothelial nitric oxide synthase; ROS = reactive oxygen species; VSMCs = vascular smooth muscle cells; ECs = endothelial cells; PEG = polyethylene glycol; * The number of animals used in animal experiments.

**Table 4 biomedicines-12-01504-t004:** Nanoparticles based on active delivery (receptor–ligand binding).

Nanocarrier	Cargo	Target	Ligand	Half-Life (h)	Experiments			Reference
					Models	Sample Size *	Outcomes	
Lipid/iron oxide	Sirolimus	VCAM-1 (ECs)	VHPKQHR peptide	13.84	Apoe-/- mouse atherosclerotic model; MAECs	n = 30	T2 relaxation time reduced by 2.7 times	[68]
Lipid	Cyclopentenone prostaglandins	VCAM-1 (ECs)	Anti-VCAM-1 antibody	NA	LDLr-/- mouse atherosclerotic model; rat peritoneal macrophages; U937	n = 6	Reduced the thickness of aortas by 32%	[69]
Lipid	Fumagillin	ανβ3-integrin (ECs)	ανβ3-integrin antagonist	NA	Hyperlipidemic rabbits	n = 71	Reduced the neovascular signal by 50% to 75%	[70]
PEG, PCL	Pigment epithelium-derived factor	ανβ3-integrin (ECs)	cRGD peptide	NA	Apoe-/- mouse atherosclerotic model; HUVECs	n = 18	Inhibit intimal thickening and reduce plaque area	[71]
Lipid	Sivelestat	ανβ3-integrin (neutrophils)	cRGD peptide	NA	Apoe-/- mouse atherosclerotic model; HUVECs; neutrophils	NA	Reduce plaque area and stabilize plaque	[72]
PLGA, PEG	Ac2-26 (N-formyl peptide receptor 2 agonist)	Collagen IV	Collagen IV-binding peptide	NA	LDLr-/- mouse atherosclerotic model	NA	Inhibit inflammation and stabilize plaque	[73]
pNIPAM	anti-inflammatory peptide	Collagen I	Collagen I-binding peptide	NA	Human aortic ECs; human coronary artery smooth muscle cells	NA	No animal experiments	[74]
Hyaluronan	3PO (glycolysis inhibitor)	CD44 (macrophage)	Hyaluronan	0.5 and 9	Apoe-/- mouse atherosclerotic model; HUVECs	n = 10	Improves endothelial continuity and inhibit inflammation	[59]
Soy PC	NA	CD36 receptor (macrophage)	oxPCs	NA	LDLr-/- mouse atherosclerotic model; primary mouse and THP-1 derived macrophages	n = 6	1.4-fold higher accumulation in aortic lesion areas	[75]
Lipid, PEG	CaMKIIγ siRNA	Stabilin-2 (macrophage)	S2P peptide	NA	LDLr-/- mouse atherosclerotic model; HeLa-Luc, RAW 264.7 and HEK-293 cells	n = 7–9 per group, 2 groups	Stabilize plaque	[76]
PEG, superparamagnetic iron oxide	microRNA-146a	Class A scavenger receptors (macrophages and ECs)	microRNA-146a	1.89	Apoe-/- mouse atherosclerotic model	n = 54	Reduce and stabilize plaques	[77]
Lipid, PLGA	LOX-1 siRNA, atorvastatin	CD44 (ECs); apolipoprotein A-I (macrophage)	apolipoprotein A-I, hyaluronic acid	NA	HUVECs; THP-1 derived macrophages	n = 66	Reduce 39% plaque size, 63% lipid accumulation, and 68% CD68+ macrophage content	[78]

Abbreviations: MAECs = mouse aortic endothelial cells; HUVECs = human umbilical vein endothelial cells; VCAM-1 = vascular cell adhesion molecule-1; ECs = endothelial cells; PEG = polyethylene glycol; PCL = polycaprolactone; cRGD = cyclic arginine–glycine–aspartic acid; PLGA = poly lactic-co-glycolic acid; pNIPAM = poly(N-isopropylacrylamide); PC = phosphatidylcholine; oxPCs = oxidized phosphatidylcholines; LOX-1 = oxidized LDL receptor-1. * The number of animals used in animal experiments.

**Table 5 biomedicines-12-01504-t005:** Drug-loaded nanoparticles delivered by intimal route.

Nanocarrier	Cargo	Delivery Method	Device Material	Coating Technology	Experiments			References
					Models	Sample Size *	Outcomes	
Chitosan, PLGA	NA	Stent	Stainless-steel	Cationic electrodeposition	Porcine coronary artery model; human coronary artery SMCs	n = 43	Comparable levels of injury, inflammation, and neointimal formation	[100]
Polyester-based polymers	Sirolimus	Balloon	NA	Drug priming after laser drills	Porcine coronary artery model	n = 130	Reduce stenosis in formulation-treated sites	[101]
Phospholipid	Sirolimus	Balloon	NA	NA	Rabbit iliac arteries model	NA	Diffusion from intima to adventitia	[102]
PLGA	miRNA-126	Stent/Balloon	NA	Electrostatic coating	Rabbit iliac arteries model; HUVECs	10 arteries per group, 2 groups	Inhibit neointimal formation	[103]
TiO_2_ nanotubular	NA	Stent	Titanium	NA	Rabbit restenosis model	n = 14	Accelerate restoration of a functional endothelium and reduce 30% stenosis	[104]
TiO_2_ nanotubular	Ag	Stent	Titanium	Electrochemical anodization and UV irradiation	Rabbit extracorporeal circulation model; rat abdominal aorta model	NA	33% decrease in the cross-sectional area of the hyperplastic tissue	[105,106]
PLGA	Docetaxel; SZ-21 (platelet IIb/IIIa receptor antibody)	Stent	Stainless-steel	Coaxial electrospray process	Bama minipigs; HUVECs and HUASMCs	n = 20	Inhibit thrombosis and in-stent restenosis	[107]

Abbreviations: SMCs = smooth muscle cells; PDMS = polydimethylsiloxane; PLGA = poly lactic-co-glycolic acid; PTA = percutaneous transluminal angioplasty; ATX = atherectomy; HA = hyaluronic acid; SA = sodium alginate. * The number of animals used in animal experiments.

**Table 6 biomedicines-12-01504-t006:** Drug-loaded nanoparticles delivered by tunica media and adventitial route.

Nanocarrier	Cargo	Delivery Method	Device Material	Coating Technology	Experiments			Reference
					Models	Sample Size *	Outcomes	
PDMS	Paclitaxel	Microneedle balloon	NA	UV cured bonding	Rabbit iliac arteries atherosclerosis model	n = 12	Greater patency and inhibit immunity	[111]
NA	Dexamethasone	Bullfrog Micro-Infusion Device	NA	NA	Patients with symptomatic peripheral artery disease receiving PTA or ATX	n = 262	Prevent restenosis	[112]
Lipid	Sirolimus	Bullfrog Micro-Infusion Device	NA	NA	Mixed breed pigs	n = 16	Reduce neointima area and neointima area	[113]
PLGA	Sirolimus	Hydrogel adventitial injection	HA/SA	NA	Rat aortic wire injury model	NA	Inhibit intimal hyperplasia and immunity	[114]
PLGA	Sirolimus	Pluronic gel periadventitial application	Kolliphor P407	NA	Rat balloon injury model; SMCs	NA	Inhibit intimal hyperplasia	[115]

Abbreviations: SMCs = smooth muscle cells; PDMS = polydimethylsiloxane; PLGA = poly lactic-co-glycolic acid; PTA = percutaneous transluminal angioplasty; ATX = atherectomy; HA = hyaluronic acid; SA = sodium alginate. * The number of animals used in animal experiments.

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
