# Peer review of "Targeted Delivery of Nanoparticles to Blood Vessels for the Treatment of Atherosclerosis"

_biomedicines, 2024, doi:10.3390/biomedicines12071504_

Round 1

Reviewer 1 Report

Comments and Suggestions for Authors

The authors have written a review about nanomedical drug delivery systems (DDTS) towards atheroschlorosis. While overall the article is well written and clear, one major problem exists: the authors have almost completely left out the explanation of how what they are describing is for atheroschlerosis. The abstract and introductino go  into great lengths to talk about nano and their advantages (this is all very well nown and published alredy and should be ommited) but have very little referencees to their importance to the disesase. Moving on, even when the authors describe the ddts and what effects they have they are so vague about how these articles were for this disease that it seems like a generic detailing of the advantages of nanomedicine for any disease.  This must be completely revamped throughout the entirety of the article to point towards the results and how they are directly correlated to this disease specifically. Stemming from this generic background also leads to the problem that a vast majority of the citations are more than several years old... if not from the early 2000's.

Comments on the Quality of English Language

The quality of the english is high and requires minor checks

Reviewer 2 Report

Comments and Suggestions for Authors

1. The abstract is too rough and needs to be more concise, focusing on the core findings of this article. It should suggest the key messages the authors want to convey to the audience.

2. The introduction is too rough and not well-structured. 

3. Figures 1, 2, and 3 lack references from the authors. Improve the font size and image resolution.

4. Better change the table numbers 2a and 2b into  "2 and 3"

5. In tables 2a, 2b, and 3, authors used the symbol  '/'; what is the meaning of that, I suggest following the standard symbols. 

6. General suggestions: The reading flow of the sections is confusing. Most sections consist solely of results from other articles, lacking further discussion and the authors' comments on how these results impact future research. I suggest authors pay attention and solve this issue.

7. Conclusion and prospects: Write the major research gaps and discuss the future research scope.

8. Polish the language and correct the grammar/typing mistakes. 

9. I recommend the manuscript for publication with the above-mentioned suggestions. 

Comments on the Quality of English Language

Polish the language and correct the grammar/typing mistakes. 

Reviewer 3 Report

Comments and Suggestions for Authors

The manuscript “Targeted Delivery of Nanoparticles to the Blood Vessels for Treatment of Atherosclerosis” is written well and effectively explores the present state of use of Nanoparticles Treatment of Atherosclerosis. There are some suggestions.

1. Authors must check all the units, typos, abbreviations, etc, and improve them. Similarly, authors must check these mistakes throughout the manuscript.

2. Authors should clearly mention the objective of present review in abstract.

3. The reference style in the text and Reference list is not as per the journal requirements. Authors carefully correct them.

4. Check for uniformity in using abbreviations.

5. What’s the source of figure. Is there any copyright issue? If taken from previously published article then authors should take copy right permission and cite it properly.

6. Remove references from conclusion.

7. Author should state the biocompatibility, toxicity, pharmacokinetic and their distribution and fate of nanoparticles.

8. Is there any nanoparticles approved by FDA for the treatment of Atherosclerosis.

9. Is there any clinical trial going on. Pls enlist. 

Comments on the Quality of English Language

Minor english correction 

Reviewer 4 Report

Comments and Suggestions for Authors

The manuscript written by Zong et al. summarizes the use of nanotechnology approaches to treat atherosclerosis. Specifically, the authors have focused this manuscript on describing the effectiveness of a myriad of novel drug delivery systems in the treatment of this disease. Overall, the manuscript is interesting, the topic is highly topical, and it clearly fits within the journal’s scope. The manuscript is well organized, and the examples selected by the authors have been clearly discussed. It is worth mentioning that the authors have included some tables that facilitate the readership of the potential of the therapeutic strategy as well as a final section that covers the potential challenges of nanomedicine for the treatment of atherosclerosis. There is a couple of comments/suggestions that the authors should consider in the revision process:

(1)    Tables 2, 3. The authors should define what “sample size” is in a table footnote. In addition, they should include additional information about the most significant outcomes of each selected example in a new column.

(2)    Please, consider the use of theragnostic approaches for the treatment of atherosclerosis in this manuscript.

(3)    The introduction section is rather limited and poor. Please, include more information about atherosclerosis pathophysiology as a new subject in this section.

Round 2

Reviewer 1 Report

Comments and Suggestions for Authors

The authors have done an incredible amount of work to revamp this article to focus on atherosclorosis. The improvements make it not only easier to read but also improve the relavance in the field. 

Comments on the Quality of English Language

The english is overall very good and just needs minor checks during proofs.

Author Response

Comments 1: The authors have done an incredible amount of work to revamp this article to focus on atherosclorosis. The improvements make it not only easier to read but also improve the relavance in the field.

Response 1: Thank you for the suggestion. We appreciate that you recognize our efforts!

Comments 2: The english is overall very good and just needs minor checks during proofs.

Response 2: Thank you for your attention to the language of our paper, and we will check carefully during proofs for minor language errors and correct them.

Reviewer 3 Report

Comments and Suggestions for Authors

Author should provide all six clinical trial in revised MS. 

Author Response

Comments 1: Author should provide all six clinical trial in revised MS.

Response 1: Thank you for your advice. We added a table (Table 2) to the paper summarizing the six clinical trials we searched. We hope you're pleased with our modifications.